# Simultaneously Enhancing the Strength, Plasticity, and Conductivity of Copper Matrix Composites with Graphene-Coated Submicron Spherical Copper

**DOI:** 10.3390/nano12061025

**Published:** 2022-03-21

**Authors:** Yulong Yang, Yilong Liang, Guanyu He, Pingxi Luo

**Affiliations:** 1College of Materials and Metallurgy, Guizhou University, Guiyang 550025, China; mryyl252@163.com (Y.Y.); hgy2726220@163.com (G.H.); lpx15186160957@163.com (P.L.); 2Key Laboratory for Mechanical Behavior and Microstructure of Materials of Guizhou Province, Guiyang 550025, China; 3National & Local Joint Engineering Laboratory for High-Performance Metal Structure Material and Advanced Manufacturing Technology, Guiyang 550025, China

**Keywords:** graphene, Cu matrix composites, interface optimization, strength plastic product, electrical conductivity

## Abstract

In this study, Cu matrix composites reinforced with reduced graphene oxide-coated submicron spherical Cu (SSCu@rGO) exhibiting both high-strength plastic product (UT) and high electrical conductivity (EC) were prepared. SSCu@rGO results in the formation of Cu_4_O_3_ and Cu_2_O nanotransition layers to optimize the interface combination. In addition, as a flow carrier, SSCu@rGO can also render graphene uniformly dispersed. The results show that SSCu@rGO has a significant strengthening effect on the Cu matrix composites. The relative density (RD) of the SSCu@rGO/Cu composites exceeds 95%, and the hardness, UT, and yield strength (YS) reach 106.8 HV, 14,455 MPa% (tensile strength (TS) 245 MPa, elongation (EL) 59%), and 119 MPa; which are 21%, 72%, and 98% higher than those of Cu, respectively. Furthermore, EC is 95% IACS (International Annealed Copper Standard), which is also higher than that of Cu. The strength mechanisms include transfer load strengthening, dislocation strengthening, and grain refinement strengthening. The plastic mechanisms include the coordinated deformation of the interface of the Cu_4_O_3_ and Cu_2_O nanotransition layers and the increase in the fracture energy caused by graphene during the deformation process. The optimized EC is due to SSCu@rGO constructing bridges between the large-size Cu grains, and graphene on the surface provides a fast path for electron motion. This path compensates for the negative influence of grain refinement and the sintering defects on EC. The reduced graphene oxide-reinforced Cu-matrix composites were studied, and it was found that the comprehensive performance of the SSCu@rGO/Cu composites is superior to that of the rGO/Cu composites in all aspects.

## 1. Introduction

Cu is widely used because of its excellent electrical conductivity (EC), but its application range is limited due to its poor mechanical properties. Therefore, preparing Cu with high EC, high strength, and high plasticity is an urgent requirement for scientific development. The introduction of strengthening phases to prepare Cu matrix composites (CuMCs) is an effective method to achieve this goal. Graphene (Gr), a two-dimensional carbon nanomaterial discovered in 2004, has received much attention owing to its excellent electrical, thermal, and mechanical properties [1,2] and is an ideal reinforcing material for metal matrix composites [3,4,5]. However, despite the success of Gr as a reinforcing phase in polymers [6,7] and ceramics [8,9], its application to metals is hampered by severe agglomeration and poor interfacial bonding with metal matrices [10]. This limits the application of Gr, especially for Gr-reinforced CuMCs, where the use of Gr often results in an enhanced strength at the cost of a reduced plasticity or EC [11,12,13,14,15]. For example, Shi et al. [16] constructed a Gr network (GN) in a Cu matrix via powder metallurgy. The reinforcing and toughening effects of GN in GN/Cu composites are emphasized. Through the construction of GN, effective grain refinement strengthening, dislocation strengthening, and load transfer strengthening are achieved. The yield strength (YS) and tensile strength (TS) of its 1.0 vol % GN/Cu composites are 128.6 MPa and 288.6 MPa, respectively, which are 81.6% and 36.8% higher than those of pure Cu, respectively. However, the plasticity is 12.0% lower than that of pure Cu. Guo et al. [17] reported a composite reinforcement phase with a pea-pod structure prepared via in-situ synthesis, which significantly improved the interface structure of the composites. The YS of the prepared Cu@GNPs@Al_2_O_3_/Cu composites can reach 300 MPa, which is 49% higher than that of pure Cu. However, its plasticity is reduced, and the elongation (EL) is only 18%. The EC is also 3.5% IACS (International Annealed Copper Standard) lower than pure Cu. Gr/Cu composites were prepared by Yang et al. [18,19] via molecular level mixing and spark plasma sintering (SPS). These composites exhibited enhanced strength and plasticity, but their EC value was lower than 80% IACS. Yang et al. [20] used wheat flour as a solid carbon source to generate Gr-coated Cu (Gr@Cu) composite powder in situ as a strengthening phase, and prepared Gr@Cu Cu composites by SPS. The results show that this method can not only obtain high-quality Gr layers, but also avoid the agglomeration of Gr and achieve strong interfacial bonding between Gr and Cu matrix. Therefore, the TS and YS of the prepared composites are 23% and 110% higher than those of pure Cu, respectively, but the plasticity and EC are reduced. Mathew [21] used the pulse electrodeposition method to prepare Cu–Gr composite foils and controlled the grain size and preferred growth direction using the current density. With this method, the author could increase the strength without reducing EC, but the plasticity was reduced.

Although many techniques and methods have been adopted, most of them focus on improving the strength of Gr/CuMCs [22,23,24], and it is difficult to simultaneously obtain high strength, plasticity, and conductivity. It is challenging to effectively combine and disperse uniformly Gr and Cu, even though the ball milling (BM) treatment can promote the physical adhesion between Gr and the Cu matrix to a certain extent. However, the high-energy used in the BM process can cause a serious damage to the structure of Gr, while low-speed BM cannot provide an effective adhesion. The use of metal nanoparticles to modify Gr can effectively solve the problem of poor bonding between Gr and the metal interface; however, the process involves complicated chemical reactions and uncontrollable loading, and the large number of impurities attached to the chemically reduced nanoparticles affects the performance of the composites. In situ growth of Gr is difficult to control the growth uniformity. The overgrown part is equivalent to the agglomeration of Gr, which becomes a defect and affects the properties of the composites.

Therefore, in this study, we take advantage of the small size and good fluidity of submicron spherical Cu (SSCu), in which the small size can be used as a void filler for large-sized Cu, and the good fluidity can be used as a dispersion carrier for rGO. rGO was coated on SSCu by electrostatic adsorption to form a SSCu@rGO core–shell structure as a composite strengthening phase. In this way, SSCu can act as a carrier to transport rGO into the large-sized Cu gap. On the one hand, the formation of sintered pores is prevented, and the relative density of sintering is improved. On the other hand, it can prevent rGO from agglomerating to form defects, promote the uniform dispersion of rGO, and form a Cu*_x_*O*_y_* nanotransition layer at the interface of rGO and Cu to strengthen the interfacial bonding. Therefore, SSCu@rGO/Cu composites with excellent comprehensive properties are obtained; their strength, plasticity, and EC are especially improved compared with pure Cu. We also discuss in detail the enhancement mechanisms of strength, plasticity, and EC, especially the plasticity enhancement mechanism. By discovering the Cu*_x_*O*_y_* nanotransition layer formed at the interface between rGO and Cu, we propose the view that the coordinated deformation of the Cu*_x_*O*_y_* nanotransition layer plays a role in plastic enhancement. In addition, we also give a new reference for the mechanism of EC improvement.

## 2. Experimental

### 2.1. Materials

The specific parameters of SSCu and large-size electrolytic dendritic Cu (Cu) used in this work are shown in Table 1. Graphene oxide (GO) consists of experimental-grade ultrapure GO, and its specific parameters are shown in Table 2. The surface modifier cetyltrimethylammonium bromide (CTAB) has a purity of over 99%.

### 2.2. Powder Preparation and Molding

#### 2.2.1. Preparation of the SSCu@GO/Cu Composite Powder

The preparation process is indicated by the red marks in Figure 1. A 1% CTAB alcohol solution was prepared and ultrasonically dispersed for 1 h. SSCu was added to the CTAB solution and stirred magnetically for 2 h. After suction, SSCu was dried at 50 °C under vacuum. After surface modification, the SSCu was added to the deionized water and kept under stirring for 0.5 h to obtain a suspension of the Cu powder, which was positively charged by zeta potential measurement (pH = 7). GO was added to the deionized water for ultrasonic stripping for 2 h to obtain a uniformly dispersed few-layer GO solution, which was negatively charged by zeta potential measurement (pH = 7). The GO solution was slowly added to the SSCu suspension, and the resulting solution was stirred for 1 h to uniformly disperse GO. SSCu@GO was then prepared via electrostatic adsorption with the modified SSCu and uniformly dispersed few-layer GO solution. Cu was added, and stirring was continued for 1 h to render SSCu@GO evenly dispersed and make it fill the Cu gaps. Finally, the suction was filtered and dried at 50 °C under vacuum to obtain the SSCu@GO/Cu composite powder.

#### 2.2.2. Preparation of the GO/Cu Composite Powder

The preparation process is indicated by the black marks in Figure 1. The GO solution treated via ultrasonic stripping and the same modified positively charged Cu were stirred continuously for 2 h, filtered, and dried at 50 °C under vacuum to obtain the GO/Cu composite powder.

#### 2.2.3. Preparation of the SSCu@rGO/Cu and rGO/Cu Composites

The thermal reduction of GO and powder consolidation molding of the composite powder using hot-press sintering protected by high-purity argon (purity ≥ 99.999%). A 60 g measure of the composite powder was loaded into a graphite mold with an inner diameter of 24 mm. When the residual pressure of the vacuum hot-pressing furnace (model VVPgr-15-2000) became less than 20 Pa, high-purity argon gas was introduced from the bottom of the furnace body at a gas flow rate of 10 L/min for protection. In order to avoid the adverse effects of flowing argon on the heating and heat preservation process, the ventilation is stopped when the pressure in the furnace reaches 0.02 MPa. As the density of argon is greater than that of air and oxygen, the argon introduced from the bottom of the furnace continuously pushes the residual air and oxygen to the top of the furnace and then releases the residual air and oxygen from the top of the furnace until the pressure in the furnace is equal to the standard atmospheric pressure. The above operation was repeated three times to maintain a high-purity atmosphere protected by high-purity argon in the furnace. Subsequently, the temperature was increased from room temperature to 600 °C at a rate of 10 °C/min, and gas was then released from the top of the furnace to discharge the impurity gas released by the CTAB pyrolysis and GO reduction during the powder sintering process. The temperature was further increased to 900 °C and kept constant for 1 h to make the sintering and thermal reduction fully proceed; it was then reduced to 200 °C at a cooling rate of 10 °C/min, and room temperature was finally reached with the furnace. A pressure of 40 MPa was applied to the mold throughout the hot-pressing sintering process.

### 2.3. Characterization

The reduction and defects of GO were analyzed via Fourier-transform infrared spectroscopy (FT-IR, Nicolet iS50) and 532 nm laser Raman spectroscopy (Thermo Fisher Dxr 2Xi Microscopic Raman Imaging Spectrometer, Waltham, MA, USA). X-ray diffraction (XRD, Empyrean sharp shadow series multifunctional X-ray diffractometer) was used to characterize the phase and dislocation density of the composites. A Leica DMI5000M optical microscope (OM) was used to investigate the grain size and surface topography of the composites. Atomic force microscopy (AFM, Bruker Dimension ICON, Billerica, MA, USA) was used to measure the size and layer thickness of GO. Field-emission scanning electron microscopy (FESEM, SUPRA 40, Zeiss, Jena, Germany) was used to observe the composite powder and fracture morphology, and its energy dispersive spectrometer (EDS) function was used to detect rGO on the fracture surface. Transmission electron microscopy (TEM, Tecnai G2 F30 S-TWIN, 300 KV, Thermo Fisher) was used to characterize the microstructure and interface structures of the graphene-reinforced CuMCs.

The density of the sintered sample was measured based on Archimedes’ principle, and the relative density (RD) was calculated based on the theoretical density. The microhardness of the composites was determined using an HVS-1000 digital microhardness tester under an applied load of 0.98 N with a dwell time of 15 s. The microhardness values of each sample were measured at least five times and then averaged. The conductivity was measured via the eddy current method (Sigma 2008B/C 60 digital conductivity meter, Beijing Cap High Technology Co., Ltd., Beijing, China), and the average value was obtained from five measurements at different positions for each sample. Tensile testing was carried out using an INSTRON8802 universal testing machine (Instron, Norwood, MA, USA) at room temperature at a crosshead speed of 0.5 mm/min. For this test, the sample was prepared in a dog-bone shape (length 20 mm, width 3 mm, and thickness 1.2 mm).

## 3. Results and Discussion

### 3.1. Phase Analysis

#### 3.1.1. XRD Analysis

Figure 2a displays the XRD spectra; it can be seen that GO exhibits a noticeable diffraction peak at 2*θ* = 9°, which corresponds to the (001) crystal plane, and the crystal plane spacing is 9.83 Å. rGO exhibits a broad diffraction peak similar to that of the graphite structure at 2*θ* = 25°, which corresponds to the (002) crystal plane, and the crystal plane spacing is 3.35 Å. The shift of the diffraction peaks and the reduction in the interplanar spacing result from the reduction in the number of oxygen-containing groups in the carbon layer of GO. The peaks of Cu, the rGO/Cu composites, and the SSCu@rGO/Cu composites are all located at 2*θ* = 43.3°, 50.4°, 74.1°, 89.9° and 95.1°, which correspond to the Cu (111), (200), (220), (311) and (222) crystal planes. The diffraction peaks of rGO, Cu_4_O_3_ and Cu_2_O are not found in XRD detection, mainly because the content of rGO was too small to be detected by XRD. In addition, the Cu_4_O_3_ and Cu_2_O nanotransition layers only exist at the interface of rGO and Cu, and their thickness is very thin, so their content is also very small and cannot be detected by XRD. This also shows that there is no obvious oxidation phenomenon during the hot-pressing sintering process, which is very important for improving the mechanical and electrical properties of the composites.

#### 3.1.2. FT-IR Analysis

Figure 2b shows the FT-IR spectra of GO, the rGO/Cu composites, and the SSCu@rGO/Cu composites. It can be seen from the figure that, after the high-temperature heat preservation treatment at 900 °C, the intensity of the oxygen-containing group peaks—such as the C=O peak at 1631 cm^−1^ and the CO peak at 1060 cm^−1^ of GO—is somewhat lower, indicating that GO is reduced to rGO. After reduction, the oxygen peak of the SSCu@rGO/Cu composites is the lowest, indicating that the reduction degree is higher and that the reduction is more complete than that of the rGO/Cu composites.

#### 3.1.3. Raman Analysis

The Raman spectra shown in Figure 2c indicate the presence of graphene in the composites; the D and G peaks of GO are located at around 1350 cm^−1^ and 1585 cm^−1^, respectively. Generally, the damage degree and the number of defects in graphene are characterized by the ratio I_D_/I_G_, where I_D_ and I_G_ are the intensities of the D and G peaks, respectively. The I_D_/I_G_ values of GO, the rGO/Cu composites, and the SSCu@rGO/Cu composites are 0.87, 1.12 and 1.26, respectively. The reason why the I_D_/I_G_ value of the composites is higher than that of GO may be that GO breaks into small pieces during the process of dispersion and hot-pressing sintering. In addition, partial agglomeration or curling may occur in the case of high GO content. The I_D_/I_G_ value of the SSCu@rGO/Cu composites is higher than that of the rGO/Cu composites. This may be due to the fact that a new chemical bond is formed at the interface of the SSCu@rGO composites [25]. The discussion section of the TEM pictures of the interface structure shows that oxides are formed at the interface of Cu and rGO, which optimizes the interfacial bonding, but destroys the SP^2^ bonding of graphene, leading to an increase in the number of defects [26].

### 3.2. Microstructural Analysis

#### 3.2.1. Powder Microstructure

Figure 3a,d shows the AFM and SEM images of GO, respectively. GO lamellae are transparent as white gauze with a lamellar diameter of about 2–5 μm. Figure 3b shows the value of the GO thickness along the red line in Figure 3a; the average thickness is about 5.06 nm. Figure 3c shows Cu, which exhibits irregularly stacked dendrites with relatively uniform size; their average size is about 12 μm. Figure 3e shows the GO/Cu composite powder. It can be seen that GO does not shrink into a coating structure, but is embedded and accumulated onto the Cu surface in sheets forming GO/Cu agglomerates, which leads to the formation of holes and defects during the sintering process. Figure 3f shows SSCu, which has a regular spherical appearance and an average particle size of about 0.74 μm; these particles show no obvious tendency to agglomerate. Figure 3g shows the SSCu@GO/Cu composite powder. Figure 3h is a magnified view of the area surrounded by the red box in Figure 3g, which shows that GO and SSCu form an effective cladding structure, and GO is shrink-wrapped on SSCu, without large pieces of GO being scattered and accumulated on the composite powder. At the same time, SSCu@GO is uniformly dispersed in the large-size Cu gaps; this promotes the uniform dispersion of GO and the filling of the sintering voids during the sintering process, resulting in an improvement of the sintering density of the composites.

#### 3.2.2. Interface Microstructure

The microstructure morphology and interfacial structure of the SSCu@rGO/Cu composites were observed via TEM, and the results are shown in Figure 4. Figure 4a,b show the shape of the sintered SSCu@rGO, and the selected area electron diffraction pattern in Figure 4a1 of the area in Figure 4a shows the diffraction pattern characteristics of Cu (200), (020) and (220). Figure 4a2 shows the diffraction pattern characteristics of C ($10\bar{1}0$) and C ($\bar{1}103$). Figure 4c,d are magnified views of the Cu–rGO interfacial region; the interface was found to be tightly bound with no voids and cracks at the interface. In order to investigate the quality of the interface, energy dispersion spectrum scans were performed at the two ends of the interface in Figure 4c, namely at points A and B. The results are shown in Figure 4c1,c2: These images prove that the dark area in Figure 4c is Cu, while the white bright area is rGO. Through further research, it was found that the interface structure shown in Figure 4c,d is not due to simple mechanical adhesion, and a more effective bonding may have occurred. In order to verify this conjecture, a high-resolution TEM image of the interface between Cu and rGO was acquired, as shown in Figure 4e.

Figure 4e1 shows a fast Fourier transform (FFT) image in which two sets of diffraction spots are visible. After calibration, these two sets of spots were confirmed to be Cu [$02\bar{2}$] and Cu_4_O_3_ [$\bar{4}80$]; furthermore, [$02\bar{2}$]Cu//[$\bar{4}80$]Cu_4_O_3_, which proves that the area is the Cu/Cu_4_O_3_ interface. After calibration of the diffraction spot of the FFT image of Figure 4e2, it was confirmed to be rGO [$\bar{1}210$], proving that the area is the Cu_4_O_3_/rGO interface. Figure 4e3–e5 illustrates the inverse fast Fourier transform (IFFT) diagrams; from these, the interplanar spacings of Cu (111), rGO (0002), and Cu_4_O_3_ (213) were found to be 0.21, 0.34, and 0.21 nm, respectively. Figure 4f shows the analysis of the high-resolution images at different positions, indicating that Cu_2_O is also generated at the interface. The FFT diagram in Figure 4f1 shows two sets of diffraction spots, namely the Cu_2_O [$\bar{2}20$] zone axis and C (0002) crystal plane, with (220)Cu_2_O‖(0002)C, which proves that the area is the Cu_2_O/rGO interface. Figure 4f2 illustrates the FFT diagram, which shows the intensity maxima for Cu [$02\bar{2}$] and Cu_2_O [$\bar{2}20$], with [$\bar{2}20$]Cu_2_O//[$02\bar{2}$]Cu, proving that the area is the Cu/Cu_2_O interface. The above results indicate that a chemical reaction occurs at the interface between Cu and rGO during the sintering process. The oxygen-containing groups on the surface of rGO react with Cu to form Cu_4_O_3_ and Cu_2_O, resulting in the three-layer structures of Cu/Cu_4_O_3_/rGO and Cu/Cu_2_O/rGO at the interface. The thicknesses of the transition layers Cu_4_O_3_ and Cu_2_O are about 5 nm; these transition layers strengthen the interfacial bonding.

Based on the above FFT and IFFT analysis results of the interface and the crystal plane orientation relationship, the misfit *ξ* of the interface can be estimated according to [27]
(1)$$\xi =\frac{\left|\frac{1}{2}d_{\left(h1k1l1\right)}\cos\theta -d_{\left(h2k2l2\right)}\right|}{d_{\left(h2k2l2\right)}}\times 100\%$$
where *d(hkl)* is the interplanar spacing, and *θ* is the misorientation angle. The maximum misfit *ξ*_max_ of the interface between Cu ($\bar{1}11$) and rGO (0002) is calculated to be 75%, and the minimum misfit ξ_min_ of the interface between Cu (200) and rGO (0002) is 15%. The misfit *ξ*_Cu_4_O_3__ of the interface between Cu_4_O_3_ (213) and rGO (0002) is 52%, and the misfit *ξ*_Cu_2_O_ of the interface between Cu_2_O (111) and rGO (0002) is 67.8%. It can thus be observed that *ξ*_min_ < *ξ*_Cu_4_O_3__ < *ξ*_Cu_2_O_ < *ξ*_max_, indicating that Cu_4_O_3_ and Cu_2_O formed at the interface can reduce the huge interface mismatch difference between Cu and rGO. However, the interface mismatch is still very large, and a large mismatch weakens the interfacial bonding.

Thus, interface strengthening and weakening exist at the same time, and the mutual blending makes the interface combination optimized, which can obtain strength and plasticity simultaneously [28,29]. As the excessive interfacial bonding makes the Cu_4_O_3_ and Cu_2_O nanotransition layers difficult to start and slip during the deformation process, a coordinated deformation cannot be achieved, resulting in cracks extending directly through the Cu_4_O_3_ and Cu_2_O transition layers to the Cu matrix, thereby reducing plasticity. If the interface is too weak, it is difficult to achieve load transfer to increase strength. Instead, rGO becomes a defect of the composites and deteriorates the performance.

#### 3.2.3. Fracture Microstructure

Figure 5 shows the SEM image of fractures in Cu, the rGO/Cu composites, and the SSCu@rGO/Cu composites. The panels indicated by the red arrows are enlarged views of the areas in the red boxes. It can be found that rGO is distributed on the fracture surface of the composites. The rGO on the fracture surface was verified by EDS (Appendix A), and the results showed that there were obvious C peaks. As shown in Figure 5a,a1, there is a large number of dimples on the surface of the Cu fracture, which is a typical plastic fracture. Figure 5b,b1,c,c1 show the rGO/Cu composites with an rGO content of 0.3 wt % and 0.5 wt %. There are a large number of pores on the fracture surface, rGO is distributed in the pores, and the size of the pores increases with the increase in the rGO content. This is due to the fact that, when rGO is directly added to the Cu matrix, due to the interface mismatch between rGO and Cu and the multidirectionality of the rGO orientation, rGO and large-size Cu cannot form an effective coating structure. rGO is embedded in sheets and randomly adsorbed onto the Cu surface to form agglomerates, which leads to the formation of pores and defects during the sintering process. This in turn reduces the RD of the composites and affects their comprehensive performance. Figure 5d,d1 shows the fracture of the SSCu/Cu composites. The surface dimples are large and deep. The strength plastic product (UT = TS × EL, EL = (L_1_ − L_0_)/L_0_, where L_1_ (mm) is the gauge length after breaking, L_0_ (mm) is the gauge length before breaking) is improved compared with that of Cu. As SSCu has the function of filling the sintering voids and refining the grains, the mechanical properties are optimized. Figure 5e,e1 show the SEM image of the SSCu@rGO/Cu composites with an rGO content of 0.3 wt %. The uniform dispersion of rGO on the fracture surface shows that SSCu@rGO effectively inhibits the agglomeration of rGO, and rGO assumes different forms on the dimples: some rGO is embedded in the dimples, some is torn, and some falls on the surface of the dimples after being pulled out. This shows that rGO is debonded and pulled out during the stretching process, preventing the cracks from propagating through the matrix. At the same time, the high length-to-width ratio of rGO and the large contact area with the matrix can also hinder crack propagation during motion [30], which causes the cracks to deflect. The above synergistic effect improves the strength and plasticity of the SSCu@rGO/Cu composites. Figure 5f,f1 show the SEM image of the SSCu@rGO/Cu composites with an rGO content of 0.5 wt %. The increase in the GO content causes the coating thickness of SSCu@GO to increase, which affects the flow of SSCu@GO. After the coating is completed, there is still a large amount of GO residue, which hinders the flow of SSCu@GO, resulting in the accumulation and agglomeration of SSCu@rGO. The agglomerates cause sintering to form large pores and defects, which reduces the performance of the composites.

### 3.3. Mechanical Properties and Electrical Conductivity

#### 3.3.1. Effect of Different SSCu Contents on the Properties of the Composites

In order to obtain CuMCs with the best comprehensive performance, the effect of different SSCu contents on the performance of the composites was explored. The performance of the composites with SSCu mass fractions of 10, 30, and 50 wt % was studied. The results are shown in Figure 6a,b. When the mass fraction of SSCu is 50 wt %, the performance of the composites is the best. The RD reaches 99%, the microhardness is 88.2 HV, and the TS and elongation (EL) are 232 MPa and 50%, respectively. In particular, the UT is 38% higher than the UT value of Cu. However, EC is reduced to 82% IACS. The addition of SSCu effectively fills the voids between the large-sized Cu, preventing the formation of sintered voids, thus improving the RD. According to the Hall–Petch relationship, the improvement of the mechanical properties is due to the refinement of the grains. The average SSCu grain size of 0.74 μm reduces the average grain size of the composites, and the greater the mass fraction of SSCu, the greater the degree of refinement. These considerations explain the performance improvement of the composites. The decrease in EC is mainly due to the increase in the number of the interfaces caused by grain refinement, which leads to an increase in the barrier to electric conduction. Based on the above results, in order to obtain a CuMCs with simultaneous excellent mechanical and electrical properties, different GO contents were coated onto SSCu with a mass fraction of 50 wt % to prepare the SSCu@rGO/Cu composites.

#### 3.3.2. Comparison of the Performance between the rGO/Cu and SSCu@rGO/Cu Composites

Figure 7 compares the properties of the SSCu@rGO/Cu and rGO/Cu composites with different mass fractions of rGO. As shown in Figure 7, the quantities describing the properties of the rGO/Cu composites exhibit a downward trend with the increase in the rGO content, and the decline is large. The RD of the SSCu@rGO/Cu composites is higher than that of Cu and the rGO/Cu composites, because the addition of SSCu@rGO fills the large-sized Cu voids and prevents the agglomeration of rGO, thereby preventing the formation of sintered pores. The other properties first increase and then decrease upon increasing the rGO content, decreasing to the lowest value later, it can basically be at the same level as Cu. As shown in Figure 7a, with the increase in the rGO content, the RD of the rGO/Cu composites decreases sharply. When the mass fraction of rGO is 0.5 wt %, the RD is only 79%. The lower RD of rGO/Cu composites is mainly due to the massive agglomeration of rGO, which forms pores and defects during the sintering process, thus leading to RD decreases. The reduction in RD has a serious impact on the performance of the composites. Although the RD of the SSCu@rGO/Cu composites also shows an overall downward trend, it retains a value of more than 95%, which ensures a superior performance for these composites. In Figure 7b, the hardness values of pure Cu, rGO/Cu composites, and SSCu@rGO/Cu composites are compared. Since the hardness value is affected by the surface of the sample, in order to avoid errors, each sample is measured five times at different positions and the average value is taken as the final hardness value. The specific measurement data are in Appendix A. The results show that the hardness value of rGO/Cu composites basically shows a decreasing trend with the increase in rGO content, especially when the mass fraction of rGO is 0.5 wt %, the microhardness of rGO/Cu composites is only 47.7 HV, which is 43% lower than that of pure Cu. This is because rGO/Cu composites do not have SSCu@rGO as a transport carrier, which leads to a large number of agglomerates of rGO at the grain boundaries, and the agglomerates become defects and holes that reduce the hardness (Appendix A) [31,32]. The hardness of SSCu@rGO/Cu composites is increased compared to Cu are up to 106.8 HV, which is 21% higher compared to pure Cu. Because SSCu@rGO promotes the uniform dispersion of rGO (Appendix A) and avoids the agglomeration of rGO, the uniformly distributed rGO at the grain boundaries can delimit the grains effectively [31,33], which achieves the effect of refining the grains and makes the hardness increase. As shown in Figure 7c, the EC value of the rGO/Cu composites decreases drastically with the increase in the rGO content, down to 73% IACS. It has been shown in Figure 6a that—when the content of SSCu is 50%—the mechanical properties of the SSCu/Cu composites are excellent, while their electrical properties are poor. However, from the results shown in Figure 7c, it is found that the addition of SSCu@rGO improves the EC value of the composites containing SSCu, which reaches a maximum of 95% IACS (this is slightly higher than that of Cu). The stress–strain curve in Figure 7d shows that the tensile performance also exhibits the same trend. The specific tensile performance data are provided in Table 3. The best reinforcement effect of the SSCu@rGO/Cu composites is 14,455 MPa%, which is higher than that of Cu by 72%. In addition, the YS increases by 98%, from 60 MPa for Cu to 119 MPa for the SSCu@rGO/Cu composites. However, with the increase in the rGO content, the UT of the rGO/Cu composites decreases, and the minimum is only 1520 MPa%, which is 82% lower than that of Cu.

The main reason for the decline in the performance of the rGO/Cu composites is that large-size Cu has sintering voids itself, and the agglomeration of rGO aggravates the formation of pores and defects. The guarantee of the excellent performance of SSCu@rGO/Cu composites is that, on the one hand, SSCu@rGO promotes the uniform dispersion of rGO and fills the sintering holes during the sintering process. On the other hand, the effective coating forms Cu_4_O_3_ and Cu_2_O at the interface to optimize the interfacial bonding. However, as shown in the schematic diagram of Figure 8b, as the GO content increases, the coating thickness of SSCu@GO will become larger, and there will still be a large amount of free GO after the coating is completed, which hinders the flow of SSCu@GO, resulting in the accumulation and agglomeration of SSCu@rGO. The agglomerates cause sintering to form pores and defects, which reduces the performance of the composites.

In this study, CuMCs with both high UT and high EC were achieved. In Figure 8a, the UT and EC values of the SSCu@rGO/Cu composites prepared in this paper are compared with those of other carbon-containing CuMCs [34,35,36,37,38,39,40,41,42,43,44,45,46,47,48,49,50,51]. It can be seen that most studies could not achieve both high UT and high EC, mainly because of the incompatibility between the strengthening and conductive mechanisms of carbon-based CuMCs. Grain refinement and load transfer can increase the UT of the composites, but they will also increase the energy barrier to electron motion, resulting in a decrease in EC. The SSCu@rGO prepared in this work via electrostatic adsorption optimizes interfacial bonding. This SSCu@rGO bridges the gap between large-sized Cu grains, and Gr on the surface provides a fast path for electric conduction. These phenomena can compensate for the negative effects of grain refinement and the interface barriers on EC.

### 3.4. Strengthening Mechanism

#### 3.4.1. Strength

In order to analyze the strengthening mechasnism (including strength, plasticity, and electrical conductivity) the composites, Cu, and the SSCu@rGO/Cu composites with an rGO mass fraction of 0.3 wt % were studied.

The main reasons for the influence of SSCu@rGO on the strength of the CuMCs are as follows: (1) The SSCu@rGO leads to an increase in the dislocation density; (2) It hinders the migration of grain boundaries and refines grains; (3) It transmits stress and bears the load during deformation. When considering only the influence of the dislocation density on the lattice distortion of the composites, the following expression [52] can be used to calculate the dislocation density *ρ* of the CuMCs:(2)$$\rho =14.4\frac{\varepsilon^{2}}{b^{2}}$$
where *ε* is the microstrain of the material, and b is the Burgers vector of Cu (0.256 nm) [29]. The value of *ε* is calculated from the XRD spectrum of the composites by combining Equation (3) with the following expression from the Williamson–Hall method [53]
(3)$$B\cos\theta =\varepsilon \;(\sin\theta )+\frac{k\lambda }{d}$$
where *B* is the full width at half maximum of the XRD peak, *θ* is the Bragg angle, k = 0.89 is a constant, and *λ* = 0.154 nm is the incident X-ray wavelength. As shown in Figure 9a,b, according to Equation (3), the point distribution of Cu and the SSCu@rGO/Cu composites with *B*cos*θ* as the *y*-axis and 4sin*θ* as the *x*-axis is linearly fitted, and the slope of the fitted straight line yields the value of *ε*. Finally, the dislocation densities of Cu and the SSCu@rGO/Cu composites were calculated to be 4.56 × 10^14^ and 5.21 × 10^14^/m^2^, respectively, which proves that the SSCu@rGO/Cu composites have a relatively high density of dislocations. The reason may be that the Cu_4_O_3_ and Cu_2_O nanotransition layers formed at the interface have the ability to coordinate the deformation, which leads to an increase in the dislocation density. This high dislocation density enhances the strength of the composites. The dislocation strengthening *σ_D_* caused by the increase in the dislocation density can be quantitatively calculated as [54]
(4)$$\sigma_{D}{=\alpha \mathrm{Gb}\rho }^{1/2}$$

The increase in the YS Δ*σ_D_* of the composites relative to the matrix caused by the increase in the dislocation density can be expressed as
(5)$${∆\sigma }_{D}{=\alpha \mathrm{Gb}(\rho }_{C}^{1/2}-{\;\rho }_{M}^{1/2})$$
where *α* = 1.25 is a constant [54], *G* is the shear modulus of the Cu matrix (4.21 × 104 MPa [55]), and the subscripts *C* and *M* denote the composites and the matrix, respectively. It was calculated that Δ*σ_D_* is about 19.8 MPa. 

SSCu@rGO fills the gaps between large-size Cu grains, which hinders the migration of grain boundaries and limits the growth of grains, so the grains are refined. The strengthening *σ_G_* caused by grain refinement can be described using the Halle–Petch relationship [56], which is expressed as
(6)$$\sigma_{G}{=\sigma }_{0}+\frac{K}{\sqrt{d}}$$

The YS increment Δ*σ_G_* of the composites relative to the matrix caused by the grain refinement is
(7)$${\Delta \sigma }_{G}=\frac{K}{\sqrt{d_{C}}}\;-\;\frac{K}{\sqrt{d_{M}}}$$
where *K* = 140 MPa μm^1/2^ is the Hall–Petch coefficient of Cu [41]. As shown in Figure 9c,d, the grain sizes of Cu and the SSCu@rGO/Cu composites were measured, and the average grain sizes are *d_M_* = 8.6 μm and *d_C_* = 4.5 μm, respectively. The grain refinement rate reaches 52%. By substituting these values into Equation (7), Δ*σ_G_* due to grain refinement is about 18.2 MPa.

The results in Figure 4e and Figure 5e,e1 show that rGO is embedded in the Cu matrix with SSCu@rGO and reacts with the Cu interface to generate Cu_4_O_3_ and Cu_2_O to strengthen the interfacial bonding. The rGO is pulled out from the interface during the tensile process and finally tears off on the fracture surface under tensile stress, further indicating that rGO plays the role of load transfer during the tensile test. The following shear-model expression [37,57,58,59] can be used to express the yield strengthening increment Δ*σ_LT_* caused by the load transfer of rGO
(8)$${\Delta \sigma }_{\mathrm{LT}}=\frac{1}{4}{\mathrm{PV}}_{\mathrm{rGO}}\sigma_{M}$$
where *P* is the ratio of the slice diameter to the thickness of rGO, *V_rGO_* is the volume fraction of rGO, and *σ_M_* is the YS of the matrix. Figure 3b shows that the thickness of rGO is 5.06 nm, and rGO is dispersed in the composites with a SSCu@rGO structure, so its effective sheet diameter is the diameter of SSCu, which is about 0.74 μm. The volume fraction *V_rGO_* is 1.2 vol %. By substituting these values into Equation (8), Δ*σ_LT_* is calculated to be 26.3 MPa.

Based on the above theoretical calculation and analysis, the contribution of SSCu@rGO to the YS *σ_C_* of the CuMCs is defined as
(9)$$\sigma_{C}{=\sigma }_{M}{+\Delta \sigma }_{D}{+\Delta \sigma }_{G}{+\Delta \sigma }_{\mathrm{LT}}$$

By substituting the calculated values Δ*σ_D_* = 19.8 MPa, Δ*σ_G_* = 18.2 MPa, Δ*σ_LT_* = 26.3 MPa, and *σ_M_* = 60 MPa into Equation (9), a theoretical value of 124.3 MPa is obtained for *σ_C_*, which is very close to the actual value of 119 MPa obtained experimentally, indicating that the strengthening mechanism can be explained. Thus, the YS of the SSCu@rGO/Cu composites is approximately doubled compared with that of the Cu matrix owing to the synergistic effects of dislocation strengthening, grain refinement strengthening, and load transfer strengthening of the CuMCs by SSCu@rGO.

#### 3.4.2. Plasticity

The improvement of plasticity benefits from the coordinated deformation ability of the Cu_4_O_3_ and Cu_2_O nanotransition layers at the interface.

Figure 10a,b shows the change in the orientation of rGO and Cu_2_O at the interface after deformation. Specifically, Figure 10a shows (220)Cu_2_O//(0002)C, while Figure 10b shows (220)Cu_2_O⊥(0002)C, and a large number of dislocations are found in the area where the crystal plane orientation changes. Therefore, this difference in the crystal plane orientation and the number of high-density dislocations is likely to be the result of the coordinated deformation of the interface. Figure 10c shows the comparison of the XRD spectra before and after deformation, which verifies our conclusion. Before deformation, the XRD peak intensities of Cu and the SSCu@rGO/Cu composites exhibit the same trend, with the peak corresponding to the (111) crystal plane having the strongest intensity followed by the peaks corresponding to the (200) and (220) crystal planes. After deformation, the XRD measurements were conducted perpendicular to the tensile direction, and it was found that the peak intensity trend for Cu does not change. However, the peak intensity trend for the SSCu@rGO/Cu composites changes, with the peak corresponding to the (220) crystal plane having the strongest intensity followed by the peaks corresponding to the (111) and (200) planes. As is well known, for face-centered cubic Cu with the {111} <110> slip system, the <111> stretching direction is a hard orientation [60], which is not conducive to plastic deformation. This change in peak intensity is due to the distortion and rotation of the crystal lamellae caused by the coordinated deformation of the interface, which renders the (220) crystal plane preferentially oriented during the stretching deformation process, resulting in an increase in the intensity of the diffraction peak after deformation [61]. In Figure 10d, the normalized strain-hardening-rate–true-strain curves reveal that the SSCu@rGO/Cu composites possess a higher strain hardening rate (~33) and a wider range of positive strain hardening intervals during homogeneous plastic deformation compared with those of Cu. It was demonstrated that the coordinated deformation at the interface of the Cu_4_O_3_ and Cu_2_O nanotransition layers endows the SSCu@rGO/Cu composites with the capability to stabilize the strain more permanently and avoid strain localization [62,63].

SSCu@rGO also leads to an increase in fracture energy during deformation. The details of these processes are shown in Figure 11: crack deflection [64], rGO debonding and pulling out, the formation of secondary cracks, and crack pinning [65].

Figure 11a shows the macroscopic fracture diagram of the upper surface of the parallel section of the tensile specimen. The crack expands in an arc-shaped zigzag shape, increasing the total crack propagation distance. Figure 11b illustrates a macroscopic fracture view of the side height surface of the parallel section. The crack propagation direction of the side height surface is deflected by 30°, which increases the total crack propagation area. The increase in the crack propagation distance and area increases the total fracture energy consumption. The fracture energy generated by crack deflection can be calculated according to the analysis of Faber and Evans [64]
(10)$$\frac{W_{C}}{W_{M}}=1+0{.28V}_{\mathrm{rGO}}\frac{l}{t}$$
where *W_C_* is the fracture energy of the composites, *W_M_* is the fracture energy of the matrix, *V_rGO_* = 1.2 vol % is the volume fraction of rGO, *l* = 0.74 μm is the diameter of rGO, and *t* = 5.06 nm is the thickness of rGO. By substituting these values into Equation (10), the theoretical fracture energy ratio W_C_/W_M_ is calculated to be 1.5. The actual fracture energy ratio can be numerically regarded as the ratio of UT. The UT values of the composites and the matrix are 14,455 and 8385 MPa% respectively, and the ratio is 1.7, which is similar to the theoretical value. This shows that, when a propagating crack encounters rGO, the propagation direction changes when the crack bypasses rGO, which increases the total fracture area and leads to an increase in the total energy consumption.

Figure 11c shows rGO debonding and pulling out. This process consumes additional energy to increase the deformation work, thereby increasing plasticity. Figure 11d shows the distribution of secondary cracks. The process of rGO debonding and moving from the interface leads to the formation of secondary cracks. The formation of secondary cracks releases the constraints on the matrix and generates additional strain. Figure 11e shows the crack pinning phenomenon. This phenomenon can be analyzed according to the dislocation theory expression
(11)$$\sigma_{C-T}={\left(\frac{{2G\gamma }_{S}}{k_{y}\sqrt{d}}\right)}^{\frac{1}{2}}$$
where *σ_C-T_* is the yield strength, *γ_S_* is the plastic deformation work, *d* is the grain size, *k_y_* is the pinning coefficient, and *G* is the shear modulus. Since the rGO content is very small, the change in *G* can be ignored. In order to facilitate the comparison between Cu and the SSCu@rGO/Cu composites, Equation (11) is expressed as
(12)$$\frac{\gamma_{\mathrm{SSCu}@\mathrm{rGO}/\mathrm{Cu}}}{\gamma_{\mathrm{Cu}}}=\frac{k_{\mathrm{SSCu}@\mathrm{rGO}/\mathrm{Cu}}}{k_{\mathrm{Cu}}}\frac{\sigma_{\mathrm{SSCu}@\mathrm{rGO}/\mathrm{Cu}}}{\sigma_{\mathrm{Cu}}}\frac{\sqrt{d_{\mathrm{SSCu}@\mathrm{rGO}/\mathrm{Cu}}}}{\sqrt{d_{\mathrm{Cu}}}}$$

The composites exhibit the pinning effect due to rGO; thus the constant k_SSCu@rGO/Cu_ is greater than k_Cu_, that is, $\frac{k_{\mathrm{SSCu}@\mathrm{rGO}/\mathrm{Cu}}}{k_{\mathrm{Cu}}}\;\geq \;1$. Then, by substituting σ_SSCu@rGO/Cu_ = 119 MPa, σ_Cu_ = 60 MPa, d_SSCu@rGO/Cu_ = 4.5 μm, and d_Cu_ = 8.6 μm into Equation (12), one obtains $\frac{\gamma_{\mathrm{SSCu}@\mathrm{rGO}/\mathrm{Cu}}}{\gamma_{\mathrm{Cu}}}\;\geq \;1.4$. In other words, $\gamma_{\mathrm{SSCu}@\mathrm{rGO}/\mathrm{Cu}}{>\;\gamma }_{\mathrm{Cu}}$ under the same tensile conditions, pinning cracks consume energy and produce plastic energy.

In summary, crack deflection, rGO debonding and pulling out, the formation of secondary cracks, and crack pinning lead to energy consumption and deformation work during the fracturing of the SSCu@rGO/Cu composites, resulting in greater stress–strain values and a corresponding enhancement in the plasticity of the composites.

#### 3.4.3. Electrical Conductivity

Gr-reinforced CuMCs are often affected by a reduction in their electrical conductivity (EC). This is mainly due to the fact that the interface mismatch between Gr and the Cu matrix leads to interface debonding and the appearance of sintering holes, which increases the barrier to electric conduction. In this experiment, the SSCu@rGO structure manufactured via electrostatic adsorption optimizes the interfacial bonding, fills the sintering voids, and reduces porosity. Figure 12a shows that the SSCu@rGO builds bridges between the gaps of large-size Cu grains, and rGO on its surface provides a fast path for electron motion, thus restoring conductivity. The conductivity boundary of the rGO and Cu two-phase composite system is derived from the theoretical model established by Hashin and Shtrikman [66]
(13)$$E_{\mathrm{top}}{=E}_{\mathrm{Cu}}+\frac{V_{\mathrm{rGO}}}{\frac{1}{E_{\mathrm{rGO}\;}{-\;E}_{\mathrm{Cu}}}+\frac{V_{\mathrm{Cu}}}{{3E}_{\mathrm{Cu}}}}$$
(14)$$E_{\mathrm{bottom}}{=E}_{\mathrm{rGO}}+\frac{V_{\mathrm{Cu}}}{\frac{1}{E_{\mathrm{Cu}}{\;-\;E}_{\mathrm{rGO}}}+\frac{V_{\mathrm{rGO}}}{{3E}_{\mathrm{rGO}}}}$$
where E_top_ and E_bottom_ are the upper and lower bounds, respectively, of the EC of the composites; E_Cu_ and E_rGO_ are the EC values of the Cu matrix and rGO, respectively; and V_Cu_ and V_rGO_ are the volume fractions of the Cu matrix and rGO, respectively. Here, E_Cu_ = 5.4 × 10^7^ S/m, and E_rGO_ = 1.1 × 10^5^ S/m [67]. When the mass fraction of rGO is 0.3 wt %, its volume fraction is about 1.2 vol %. The volume fraction of Cu is about 98.8 vol %. By substituting these values into Equations (13) and (14), the values of E_top_ and E_bottom_ are calculated to be 5.5 × 10^7^ and 1.8 × 10^7^ S/m, respectively, which correspond to 96% and 32% IACS. The actual measured EC value is 93 ± 1% IACS, which is similar to the theoretical upper bound of the EC of the composites. This shows that the SSCu@rGO structure effectively promotes the combination of the two-phase interface, fills the sintering gap, and improves the electron motion efficiency; thus, the EC of the SSCu@rGO/Cu composites is restored to the pure-Cu level. However, as shown in Figure 12b, with the increase in rGO content, the phenomenon of multi-layer coating becomes serious, which leads to the accumulation and agglomeration of rGO on the Cu surface, and the gap between rGO sheets increases, which hinders the migration of electrons, so the conductivity decreases instead.

## 4. Conclusions

In this study, a Cu matrix composite with high UT and high EC was prepared through electrostatic adsorption and hot-pressing sintering. The main conclusions are summarized as follows:

(1)The combination of SSCu and Cu can improve the sintering RD and mechanical properties, but the EC value is reduced.(2)The rGO/Cu composites cannot provide an effective coating structure, resulting in a large amount of rGO agglomeration. The agglomerates are sintered to form pores and defects, which reduce the RD, mechanical properties, and EC.(3)SSCu@rGO increases the sintering RD, uniformly disperses the graphene, and promotes the interface reaction between rGO and Cu to form Cu_4_O_3_ and Cu_2_O, which strengthen the interfacial bonding. The large mismatch between Cu_4_O_3_, Cu_2_O, and the interface weakens the interfacial bond. It is the balance between these strengthening and weakening effects that optimizes the interface, which can simultaneously achieve good strength and plastic coordination.(4)The RD of the SSCu@rGO/Cu composites exceeds 95%. Hardness, TS, EL, and YS can reach 106.8 HV, 245 MPa, 59%, and 119 MPa respectively, which are greatly improved compared with those of Cu. The EC value of 95% IACS is also higher than that of Cu.(5)The SSCu@rGO/Cu composites exhibit an improved strength due to dislocation strengthening, grain refinement strengthening, and load transfer strengthening. The Cu_4_O_3_ and Cu_2_O nanotransition layers possess a coordinated deformation ability during the deformation process. Furthermore, rGO increases the fracture energy during deformation via rGO debonding and pulling out, crack deflection, the formation of secondary cracks, and crack pinning, which enhance the plasticity. SSCu@rGO builds bridges between large-size Cu grains, and rGO on its surface provides a fast path for electron conduction, which restores the EC value.

## Figures and Tables

**Figure 1 nanomaterials-12-01025-f001:**
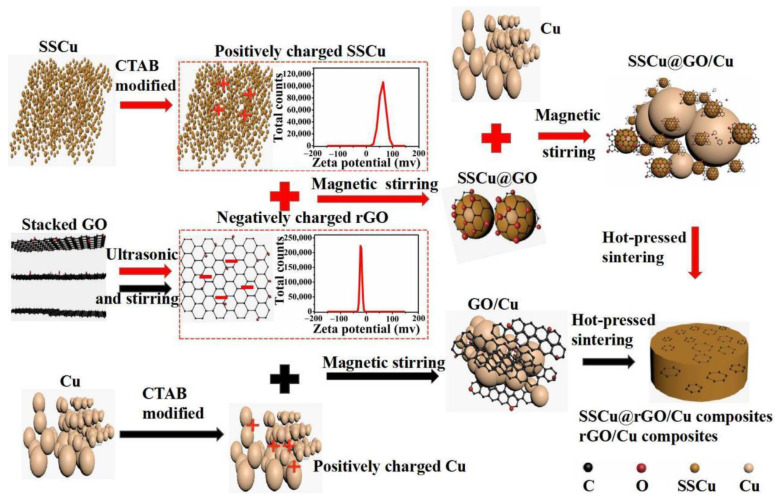
Schematic diagram of the preparation process of the SSCu@rGO/Cu and rGO/Cu composites.

**Figure 2 nanomaterials-12-01025-f002:**
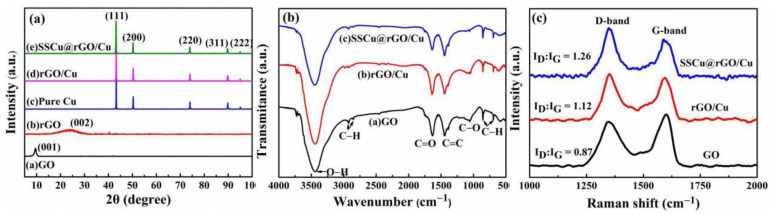
Phase analysis of GO, rGO, the rGO/Cu composites, and the SSCu@rGO/Cu composites: (**a**) XRD spectra, (**b**) FT-IR spectra, and (**c**) Raman spectra.

**Figure 3 nanomaterials-12-01025-f003:**
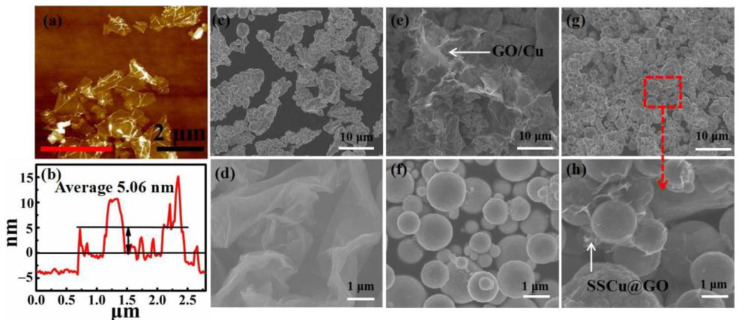
(**a**) AFM image of GO. (**b**) Thickness along the red line in panel (**a**). SEM image of (**c**) Cu, (**d**) GO, (**e**) the GO/Cu composite powder, (**f**) SSCu, and (**g**) the SSCu@GO/Cu composite powder. (**h**) Magnified view of the area surrounded by the red box in panel (**g**).

**Figure 4 nanomaterials-12-01025-f004:**
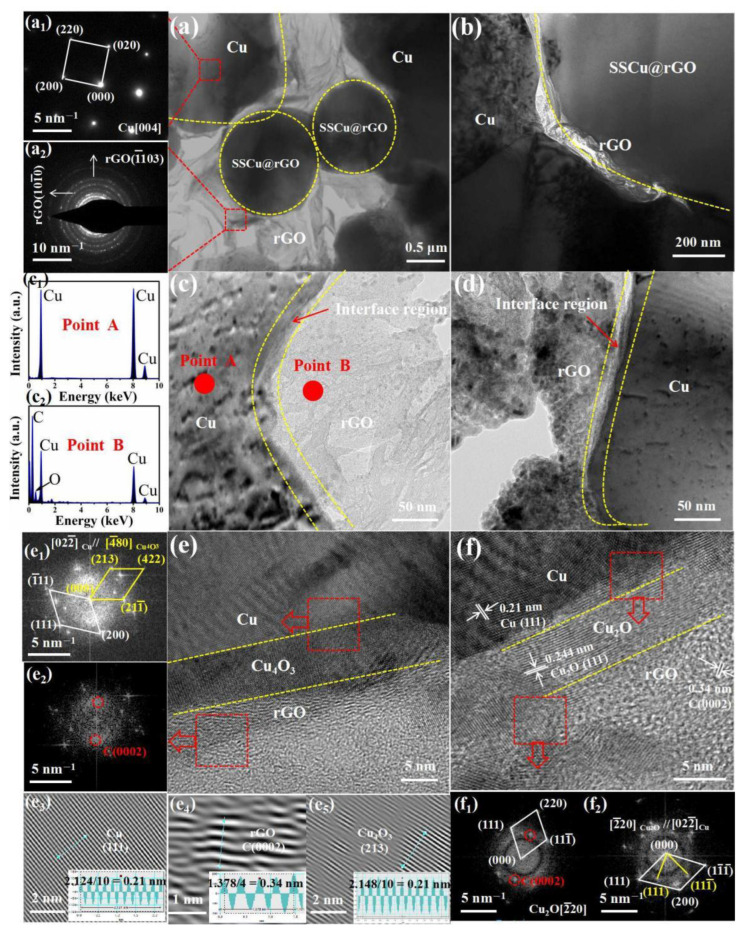
TEM images of the SSCu@rGO/Cu composites: (**a**,**a****1**,**a****2**,**b**) morphology of SSCu@rGO. (**c**,**c****1**,**c****2**,**d**) Interface structure diagram of Cu and rGO. (**e**,**e****1**–**e****5**) Interface analysis of Cu/Cu_4_O_3_/rGO. (**f**,**f****1**,**f****2**) Interface analysis of Cu/Cu_2_O/rGO.

**Figure 5 nanomaterials-12-01025-f005:**
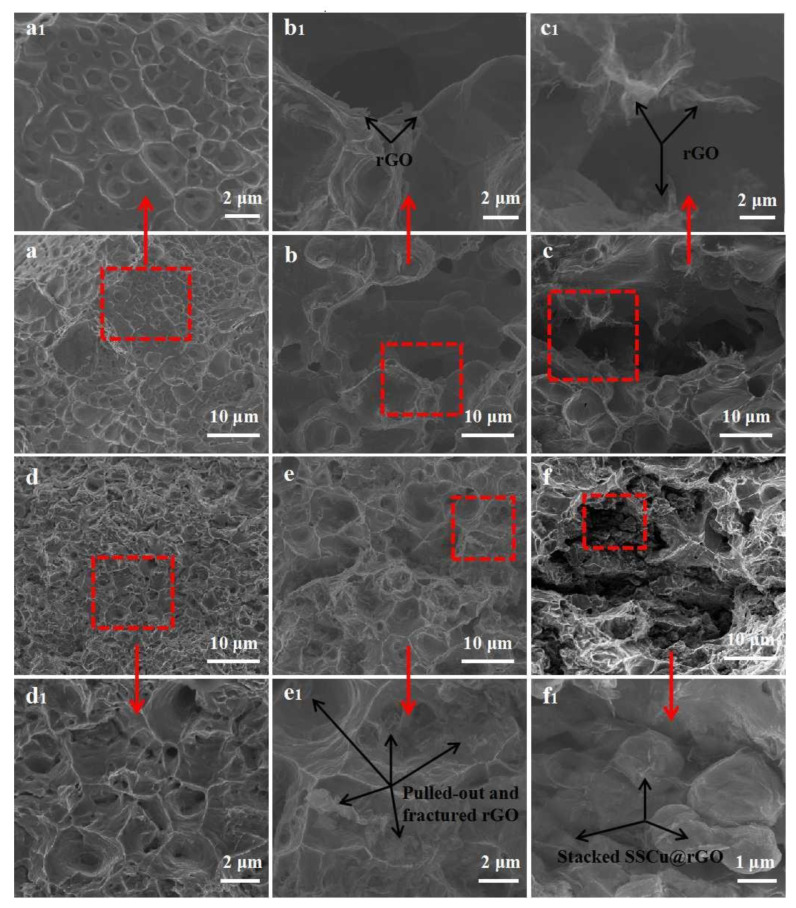
SEM images of the tensile fracture surface of the composites: (**a**) Cu, (**b**) 0.3 wt % rGO/Cu, (**c**) 0.5 wt % rGO/Cu, (**d**) SSCu/Cu, (**e**) SSCu@0.3 wt % rGO/Cu, and (**f**) SSCu@0.5 wt % rGO/Cu. (**a****1**–**f****1**) are the magnified views of the areas surrounded by the red boxes in (**a**–**f**), respectively.

**Figure 6 nanomaterials-12-01025-f006:**
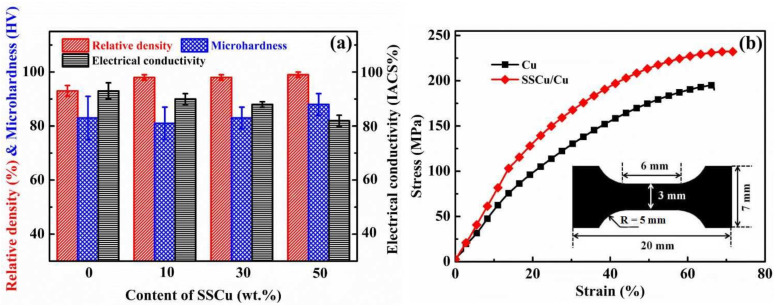
Comparison of the performance of SSCu/Cu with different SSCu contents: (**a**) relative density (RD), microhardness, and conductivity; (**b**) stress–strain curves.

**Figure 7 nanomaterials-12-01025-f007:**
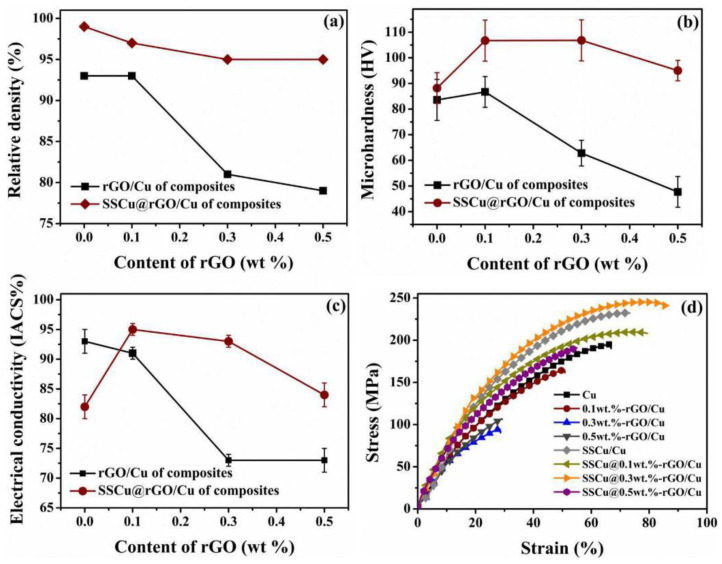
Comparison of the performance between the rGO/Cu and SSCu@rGO/Cu composites: (**a**) relative density (RD), (**b**) microhardness, (**c**) electrical conductivity, and (**d**) stress–strain curves.

**Figure 8 nanomaterials-12-01025-f008:**
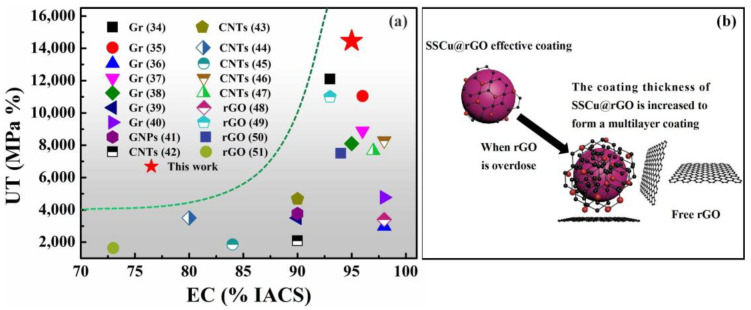
(**a**) Comparison of the UT and EC values of the SSCu@rGO/Cu composites prepared in this work with those of other carbon-containing CuMCs; (**b**) SSCu@rGO forms a multilayer coating, and there is residual rGO after the coating is completed.

**Figure 9 nanomaterials-12-01025-f009:**
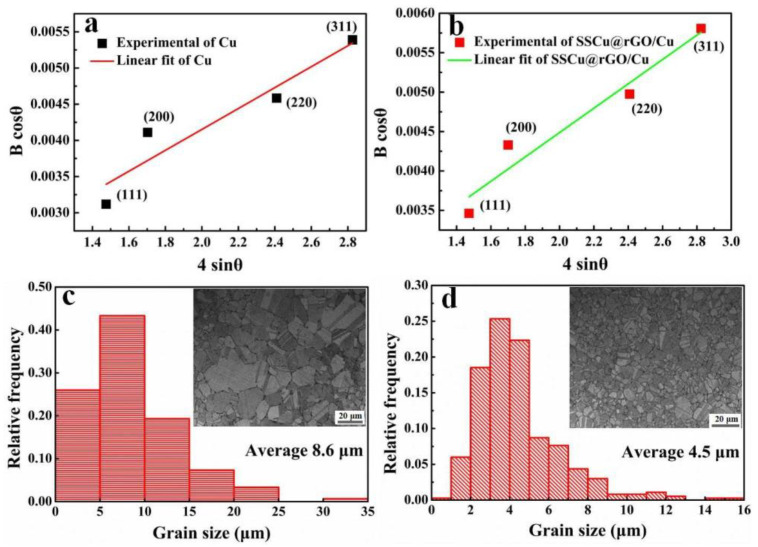
(**a**,**b**) are the linear fitting diagrams of *B*cos*θ* vs 4sin*θ* for Cu and the SSCu@rGO/Cu composites, respectively. (**c**,**d**) are the grain size statistics of Cu and the SSCu@rGO/Cu composites, respectively.

**Figure 10 nanomaterials-12-01025-f010:**
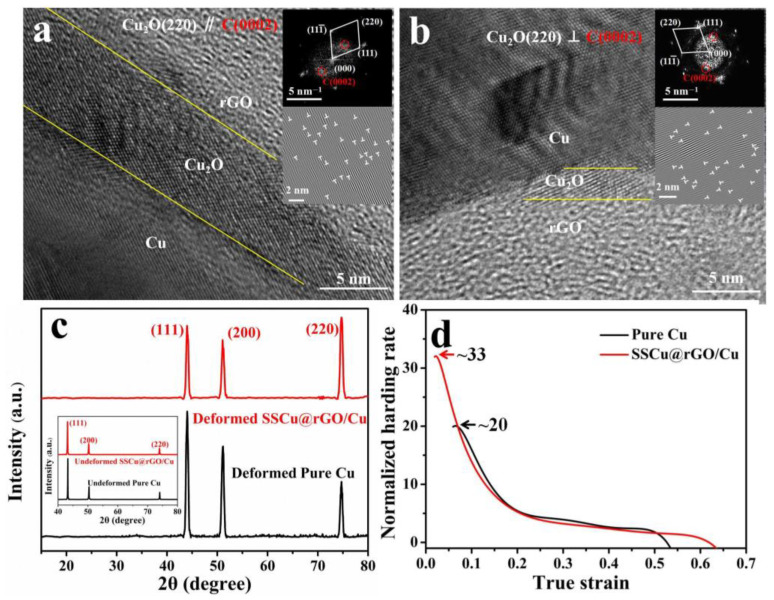
(**a**,**b**) Orientation relationship of the interface after deformation. (**c**) Comparison of the XRD patterns before deformation and perpendicular to the tensile direction after deformation. (**d**) Normalized curve of the strain-hardening rate vs. true strain.

**Figure 11 nanomaterials-12-01025-f011:**
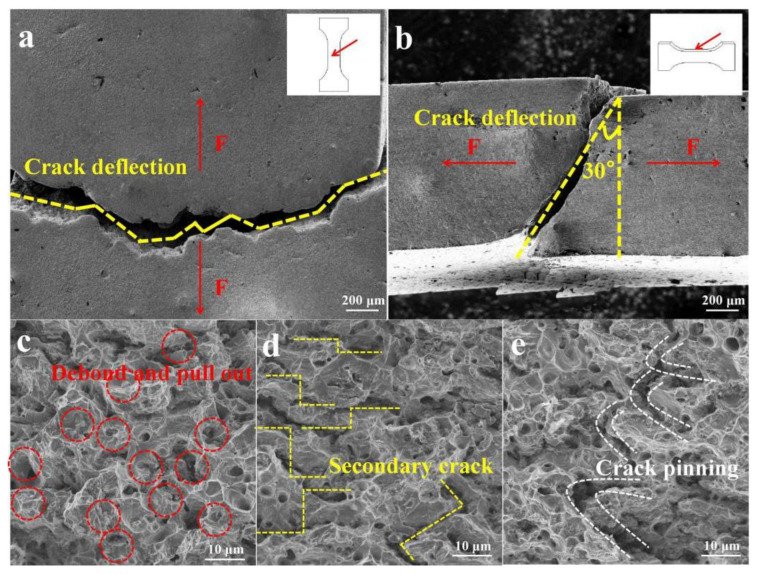
(**a**) Deflection of a crack on the upper surface of the parallel section of the specimen, (**b**) deflection of a crack on the side surface of the parallel section of the specimen, (**c**) rGO debonding and pulling out, (**d**) secondary cracks, and (**e**) crack pinning.

**Figure 12 nanomaterials-12-01025-f012:**
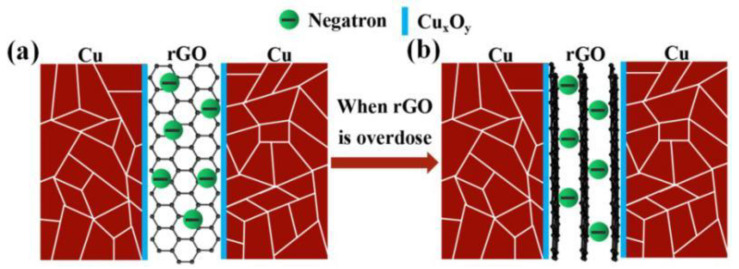
(**a**) When effectively coated, the rGO on the Cu surface provides a fast path for electron motion. (**b**) When rGO is overdosed, there is a phenomenon of multi-layer coating, which limits the movement of electrons.

**Table 1 nanomaterials-12-01025-t001:** Experimental Cu parameters.

Product	Size (μm)	Chemical Composition (%)						
		Cu	Zn	Pb	Sb	Fe	Ni	S
SSCu	0.5–1	99.95	0.001	0.001	0.001	0.042	0.005	-
Cu	~12	99.98	0.005	0.006	0.003	0.004	-	0.002

**Table 2 nanomaterials-12-01025-t002:** Experimental GO parameters.

Product	Purity	Peelable Rate	Diameter (μm)	C (%)	O (%)	S (%)
GO	99%	>96%	2–5	<45%	>50%	<1.5%

**Table 3 nanomaterials-12-01025-t003:** Tensile properties.

Samples	Tensile Strength TS (MPa)	Elongation EL (%)	Streng Plastic Product UT (MPa %)	Yield Strength YS (MPa)
Cu	195	43	8385	60
0.1%rGO/Cu	165	30	4950	70
0.3%rGO/Cu	95	16	1520	50
0.5%rGO/Cu	107	18	1926	56
SSCu/Cu	232	50	11,600	106
SSCu@0.1%rGO/Cu	210	59	12,390	81
SSCu@0.3%rGO/Cu	245	59	14,455	119
SSCu@0.5%rGO/Cu	191	31	5921	52

## Data Availability

The data presented in this study are available on request from the corresponding author. The data are not publicly available due to the confidentiality requirements of the data related to the project.

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
