# Peer review of "Simultaneously Enhancing the Strength, Plasticity, and Conductivity of Copper Matrix Composites with Graphene-Coated Submicron Spherical Copper"

_nanomaterials, 2022, doi:10.3390/nano12061025_

Round 1

Reviewer 1 Report

The manuscript investigates the physical and mechanical properties of Cu/r-GO composite prepared with hot press. The manuscript has been well organized and can be published after some revision as below:

1- the introduction should be modified based on new references, especially in the case of Cu/r-GO composite the authors can find many references.

2- it is suggested that authors explain the novelty of their work in the final paragraph of the introduction.

3- Fig. 5, it is absolutely known that detecting graphene and/or r-GO at the fracture surface of composites is very difficult. it is suggested that the authors prepare a MAP analysis (SEM) to detect r-GO properly ( it is very hard to find an appropriate EDS elemental mapping image but it can help to improve the quality of the manuscript)

4- in the case of the hardness of composite the authors should take more care about data, especially regarding the fact that hardness is dependent to the surface of the sample, and the differences between pure Cu and prepared composites' hardness should be justified in the proper way

Reviewer 2 Report

Dear Authors

It is a very interesting and well-presented work. The characterization of prepared samples from all aspects provide a great insight into the property enhancement mechanisms and why does that happen.

I wish you success in the publication of this work.

Author Response

We thank the reviewers for acknowledging our work.